# Body Weight, Central Adiposity, and Fasting Hyperglycemia Are Associated with Tumor Characteristics in a Brazilian Cohort of Women with Breast Cancer

**DOI:** 10.3390/nu14224926

**Published:** 2022-11-21

**Authors:** Clara Gioseffi, Patricia de Carvalho Padilha, Gabriela Villaça Chaves, Livia Costa de Oliveira, Wilza Arantes Ferreira Peres

**Affiliations:** 1José Alencar Gomes da Silva National Cancer Institute (INCA), Rio de Janeiro 20560-120, Brazil; 2Department of Nutrition and Dietetics, Josué de Castro Nutrition Institute, Health Science Center, Campus Fundão, Federal University of Rio de Janeiro, Rio de Janeiro 21941-902, Brazil

**Keywords:** breast cancer, abdominal obesity, central adiposity, hyperglycemia, diabetes, menopausal status

## Abstract

The aim of this study was to evaluate the association of overweight, obesity, excess central adiposity, hyperglycemia, and diabetes mellitus with tumor characteristics in breast cancer. In this retrospective cohort study that enrolled 2127 women with breast cancer, the independent variables collected were fasting blood glucose, body mass index, central adiposity (waist circumference and waist-to-hip circumference ratio (WHR)), and waist-to-height ratio. The tumor characteristics (infiltrating, ductal grade, hormone receptor-positive (HR+), human epidermal growth factor receptor, triple negative, size, lymph node involvement, and clinical stage) were the dependent variables. Most of the women were postmenopausal (73.5%), with an infiltrating tumor (83.0%), HR+ (82.0%), and overweight or obese (71.0%). For the premenopausal women, obesity was associated with grade 3 ductal tumor (odds ratio (OR): 1.70; 95% confidence interval (95% CI): 1.09–2.66), triple negative (OR: 1.37, 95% CI: 1.08–3.24), and size ≥ 2 cm (OR: 2.20, 95% CI: 1.36–3.56). For the postmenopausal women, obesity was associated with WHR, infiltrating tumor (OR: 1.73, 95% CI: 1.56–1.95), size ≥ 2 cm (OR: 1.38, 95% CI: 1.11–1.71), lymph node involvement (OR: 1.24, 95% CI: 1.02–1.56), and stages III–IV (OR: 1.76, 95% CI: 1.30–2.65). Excess body weight and central adiposity were associated with tumor aggressiveness characteristics in women with breast cancer, confirming the importance of nutritional status.

## 1. Introduction

The global incidence of cancer could reach 28.4 million in 2040, representing a 47.0% increase compared to 2020. After melanoma skin cancer, the type of cancer with the second highest incidence among women is breast cancer, which is also the second highest cause of cancer deaths among women [1]. In Brazil, the incidence of breast cancer is also the second highest among women after nonmelanoma skin tumors. Although the prognosis for breast cancer is relatively good if it is diagnosed and treated in a timely manner, adjusted and proportional breast cancer mortality rates make it the leading cause of death from the disease among women in Brazil [2].

Evidence shows that excess weight and body fat are risk factors for the incidence and worse prognosis of several types of cancer, including breast cancer [3,4]. This situation that has been exacerbated in recent decades by the global obesity epidemic, with a rising prevalence of excess body weight (overweight and obesity) around the world. In 2016, around 1.97 billion adults worldwide were overweight or obese [5]. It is estimated that by 2025, 2.3 billion adults will be overweight, including 700 million who are obese [6].

The impact of excess weight and body fat on metabolic profile and overall health of individuals is becoming increasingly evident, especially its association with breast cancer incidence and mortality [3,4]. Molecular mechanisms originating from the adipose tissue of obese individuals increase inflammation and negatively regulate antitumor immunity while promoting angiogenesis, tumor growth, and invasion of other tissues [4]. However, there is no consensus regarding the association of excess weight and body fat with tumor characteristics related to breast cancer tumor aggressiveness in women according to menopausal status [3,4,7].

The control of breast cancer, excess weight, and body adiposity is considered a public health issue given their high incidence in the world population. However, the scientific literature lacks robust analysis capable of generating evidence to support the planning of public and healthcare policies aimed at controlling these diseases. Thus, the aim of this study was to evaluate the association of overweight, obesity, excess central adiposity, fasting hyperglycemia, and diabetes mellitus with tumor aggressiveness characteristics in pre- and postmenopausal women with breast cancer.

## 2. Methods

This retrospective cohort study was conducted on women with breast cancer in the pretreatment phase who were candidates for surgery between January 2008 and December 2010 at a reference cancer control unit. The study was approved by the institution’s research ethics committee (CAAE 52969216.9.0000.5274), and the signing of a free and informed consent form was waived due to the study design.

Eligibility criteria were ≥20 years old, female, breast cancer diagnosis, no cancer treatment, and candidate for surgical treatment. Patients who were undergoing neoadjuvant chemotherapy, had absence of malignancy after histopathological confirmation, had a history of primary tumor prior to breast cancer, and had missing anthropometric and fasting blood glucose data from the hospital records were excluded.

### 2.1. Data Collection

Data were extracted from hospital records. The following variables were obtained at the time of hospital admission for surgery: age group (<60 vs. ≥60 years); menopausal status, where menopausal was equated with patient-reported amenorrhea for >12 months (premenopausal vs. postmenopausal); type of surgery (conservative vs. mastectomy); and fasting blood glucose (normal (<99 mg/dL) vs. impaired glucose tolerance (≥100 to 126 mg/dL) vs. diabetes mellitus (≥127 mg/dL)) [8].

A team of trained nutritionists from the institution conducted a routine anthropometric assessment of patients upon admission. According to the institution’s technical standard, a nonextendable tape measure was used to measure waist circumference (WC) around the midpoint between the last rib and the iliac crest at the end of expiration and hip circumference (HC) at the level of maximal protrusion of the gluteal muscles [5].

Body weight (in kg) was measured using an electronic scale (Líder, Empresa, País; 0.1 kg accuracy) with the individual standing barefoot, wearing light clothing, free of accessories, in the center of the balance platform in an erect position, and looking at a fixed point at eye level with feet together and arms extended down the body. Height (0.5 cm accuracy) was measured using a stadiometer attached to the electronic scale after weight measurement under the same conditions.

Body mass index (BMI) was calculated by dividing body weight (kg) by height squared (m^2^) (<25 kg/m^2^ (low and normal weight) vs. ≥25.0 to 29.9 kg/m^2^ (overweight) vs. >29.9 kg/m^2^ (obesity)) [5]. Central adiposity was measured using WC (<88.0 cm (normal) vs. ≥88.0 cm (elevated with substantially increased metabolic risk)) [5]. Waist-to-hip circumference ratios (WHR) were obtained by dividing WC by HC (<85 vs. >85 (high central adiposity)) [5], and the waist-to-height ratio (WHtR) was obtained by dividing WC by height (<0.5 vs ≥0.5) [9].

The dependent variables evaluated were the tumor aggressiveness characteristics, namely, infiltrating tumor (no vs. yes); grade 3 ductal (no vs. yes); hormone receptor-positive (HR+) (no vs. yes); human epidermal growth factor receptor 2 (HER-2)-positive (no vs. yes); size >2 cm (no vs. yes); triple negative (no vs. yes); lymph node involvement (no vs. yes); and clinical stage III and IV (no vs. yes).

### 2.2. Statistical Analysis

Analyses were performed using Stata 13.1 (StataCorp., College Station, TX, USA). Statistical significance was set at 5%. The analyses were performed according to menopausal status. Data were expressed as absolute frequencies (*n*) and percentages (%), categorical variables were compared using the chi-square test for proportions, and continuous variables were compared using the Mann–Whitney *U* test as they were not normally distributed according to the Kolmogorov–Smirnov test. Graphic representations were performed between the main independent and dependent variables.

Univariate logistic regression analyses were performed using the odds ratios (OR) with a confidence interval (CI) of 95%. All variables with *p*-value < 0.20 in the univariate analyses were tested in the multiple models. The forward–backward selection method was used, i.e., variables were added to the model one by one in increasing order of *p*-value, and only those with *p*-value < 0.050 were retained. This procedure was adopted for each dependent variable in the study according to the menopausal period, resulting in 16 multiple models.

## 3. Results

A total of 2127 women with breast cancer were included, mainly in the postmenopausal period (*n* = 1563; 73.5%) (Figure 1). Overall, 60.0% had mastectomy-type surgery (data not shown), 49.0% had fasting hyperglycemia and/or diabetes mellitus, 71.0% were overweight/obese, central adiposity ranged from 56.0% (WC) to 58.0% (WHR), and WHtR ≥ 0.5 was 83.0%. All these variables, except for WHtR, were higher in the postmenopausal women (*p*-value < 0.001) (Figure 2).

In the premenopausal women, 83.5% had infiltrating tumor, 83.4% were HR+, and 77.8% were HER-2-negative, and 65.4% had tumor size ≥ 2. In the postmenopausal women, these figures were 82.5%, 80.5%, 81.8%, and 62.2%, respectively (Table 1 and Table 2). Higher proportions of overweight/obesity were observed in the premenopausal women with tumor size ≥ 2 cm (plus a higher prevalence of WC ≥ 88 cm), triple negative, grade 3 ductal carcinoma, and clinical stages III and IV (plus a higher prevalence of hyperglycemia and WHtR ≥ 0.5) (*p*-value < 0.050) (Table 1). Higher proportions of overweight and/or obesity were observed in the postmenopausal women with grade 3 ductal carcinoma, triple negative, and clinical stages III and IV (*p*-value < 0.050). Furthermore, the prevalence of WHR ≥85 was higher among the postmenopausal women with tumor size ≥ 2 cm, lymph node involvement, and clinical stages III and IV (*p*-value < 0.050) (Table 2).

In the premenopausal patients, the multiple analyses showed that having a grade 3 ductal tumor was associated with obesity (OR: 1.70; 95% CI: 1.09–2.66), triple negative was associated with obesity (OR: 1.37; 95% CI: 1.08–3.24) and WHR ≥ 85 (OR: 1.53; 95% CI: 1.05–1.98); size ≥ 2 cm was associated with obesity (OR: 2.20; 95% CI: 1.36–3.56), and stages III and IV were associated with hyperglycemia (OR: 2.37; 95% CI: 1.34–4.56) and WHtR ≥ 0.5 (OR: 3.50; 95% CI: 1.14–10.77) (Table 3).

In the postmenopausal patients, the multiple analyses showed that infiltrating tumor (OR: 1.73; 95% CI: 1.56–1.95) and size ≥ 2 cm (OR: 1.38; 95% CI: 1.11–1.71) were associated with WHR ≥ 85, lymph node involvement was associated with WHR ≥ 85 (OR: 1.24; 95% CI: 1.02–1.56) and WHtR ≥ 0.5 (OR: 1.70; 95% CI: 1.06–1.97), and clinical stages III and IV were associated with overweight (OR: 1.44; 95% CI: 1.28–1.70) and WHR ≥ 85 (OR: 1.76; 95% CI: 1.30–2.65) (Table 4). Figure 3 summarizes these results, illustrating the different associations according to menopausal status.

## 4. Discussion

The highlight of this study is the simultaneous assessment of the association of different tumor characteristics with excess body weight, central adiposity, hyperglycemia, and diabetes mellitus in pre- and postmenopausal women with breast cancer. Its main results demonstrate that these independent parameters, especially obesity in premenopausal and excess central adiposity in postmenopausal women, may play a role in tumor phenotypes.

Assessing these parameters with methods that are inexpensive and widely used in clinical practice, this study found a high prevalence of excess body weight (overweight/obesity), central adiposity, and fasting hyperglycemia and/or diabetes mellitus, especially in postmenopausal patients (*p*-value < 0.001). Such evidence has been widely demonstrated in the scientific literature, and the prevalence found in our study was similar and even higher than those reported in previous studies [10,11,12].

Through multivariate analyses, our study demonstrated that in premenopausal women with breast cancer, obesity was associated with grade 3 ductal tumors, triple negative (in addition to WHR), and tumor size ≥2 cm. Moreover, the presence of fasting hyperglycemia was associated with clinical stages III and IV. In the postmenopausal period, excess central adiposity (by WHR) was associated with infiltrating tumor, tumor size ≥2 cm, lymph node involvement (in addition to WCR), and clinical stages III and IV (in addition to overweight). For postmenopausal women, these adverse factors are well established for the incidence of breast cancer as well as its characteristics and prognosis [12,13,14]. However, previous studies addressing the premenopausal period have yielded divergent results [4,7]. Different prevalence of obesity and the heterogeneity of breast cancer in different populations may contribute to the disparity of these results.

Metabolic alterations, such as insulin resistance and hyperinsulinemia, which are both common in obesity, can stimulate the growth of tumors, particularly those with a primary site located in the breast. The mechanism is unclear, but central adiposity, insulin resistance, sex steroids, adipokines, and systemic inflammation are potential factors involved [15]. Obesity can promote an increase in local and circulating proinflammatory cytokines and tumor angiogenesis and stimulate the population of cancer stem cells more likely to cause tumor growth, invasion, and metastasis, including in premenopausal women [4]. In addition, patients with obesity might have more difficulty feeling a small lesion in a relatively large breast, making it more likely for the tumor mass to be larger upon diagnosis. Thus, the high prevalence of overweight and obesity observed in our population is a possible explanation for our findings in both pre- and postmenopausal status.

Our results are supported by several findings. A previous study showed that postmenopausal breast cancer at a more advanced clinical stage at diagnosis was more prevalent in women with higher central adiposity (*p*-value = 0.007) and increased blood glucose (*p*-value = 0.047) [16]. Other studies have shown that BMI ≥ 25 kg/m^2^ increases the risk of developing triple negative tumors (*p*-value < 0.001), larger tumor size (*p*-value < 0.001), and greater lymph node involvement (*p*-value < 0.001), irrespective of menopausal status [13,17]. Contiero et al. [18] found that women with breast cancer and BMI ≥ 25 kg/m^2^ were at a significantly higher risk of recurrence and distant metastasis than those with BMI < 25 kg/m^2^. A recent systematic review of the literature demonstrated an association between obesity and triple negative breast cancer [19]. The mechanism of the impact of excess weight on the risk of occurrence of tumor aggressiveness characteristics in women in different menopausal states deserves further study.

Hougton et al. [20] found central adiposity to be positively associated with breast cancer in pre- and postmenopausal women independent of BMI, reinforcing the fact that mechanisms other than estrogen may also play a role in the relationship between central adiposity and breast cancer. Unlike our findings, which showed an association only in premenopausal women, a previous study observed greater tumor size in women with excess weight (*p*-value = 0.027) and central adiposity (*p*-value = 0.027), and this latter variable was a significant predictor of disease recurrence and death, regardless of menopausal status [21].

Besides its role in cancer promotion and tumor phenotypes, obesity has also been associated with worse clinical outcomes in cancer, such as surgical complications and greater chemotherapy toxicity [22]. Although the role of adiposity in survival is still under debate, a recent meta-analysis showed that obesity, when assessed by BMI, was associated with modestly worse disease-free survival and overall survival in all breast cancer subtypes (HR+HER2−, HER2+, and triple negative) [23]. On the other hand, imaging-measured visceral, subcutaneous, and total adiposity were not significantly associated with survival outcomes [24]. One explanation for these controversial results is not considering skeletal muscle in the analysis, which is an important predictor of cancer survival [25].

As well as obesity, a previous study also showed that fasting hyperglycemia at diagnosis was a predictor of worse prognosis in the short and long term in women with breast cancer. The risk of distant metastasis, which is important information for the clinical staging of the disease, was significantly higher for all the other quintiles compared to the lowest blood glucose quintile (reference < 87 mg/dL) (respective hazard ratios: 1.99, 95% CI 1.23–3.24; 1.85, 95% CI 1.14–3.0; 1.73, 95% CI 1.07–2.8; and 1.91, 95% CI 1.15–3.17) [18].

Limitations of this study include the fact that it was a unicentric study that did not represent the entire Brazilian population. In addition, it had a retrospective design using data collected from medical records, making direct anthropometric measurements impossible. Some other data of interest were also missing from hospital records (under-reporting), such as insulin and dietary assessment, which could provide insights into the interrelationship between diet, nutritional status, and tumor characteristics [26]. Finally, despite the limitations of using BMI as an indicator for adiposity, this index remains widely used as a practical and affordable tool [27]. However, it does not distinguish between lean and fat mass or identify the location of body fat. This is why measures of central adiposity—HC and WHR—are so important, which, together with WHtR, correlate well with abdominal obesity [28].

The strength of our study is the sample size, representing approximately 50% of the number of breast cancer diagnoses in the state of Rio de Janeiro during the study period [29]. In addition, we assessed the association between patient’s clinical features and their association with different tumor characteristics, which has been underexplored in the literature. Finally, the statistical analyses of the data were robust.

## 5. Conclusions

We conclude that different factors are associated with tumor aggressiveness characteristics, especially obesity and excess central adiposity, according to menopausal status, as evaluated by methods widely used in clinical practice. In order to help prevent tumors with aggressive characteristics, efforts should be concentrated on modifiable factors, such as nutritional status, both in pre- and postmenopausal women.

## Figures and Tables

**Figure 1 nutrients-14-04926-f001:**
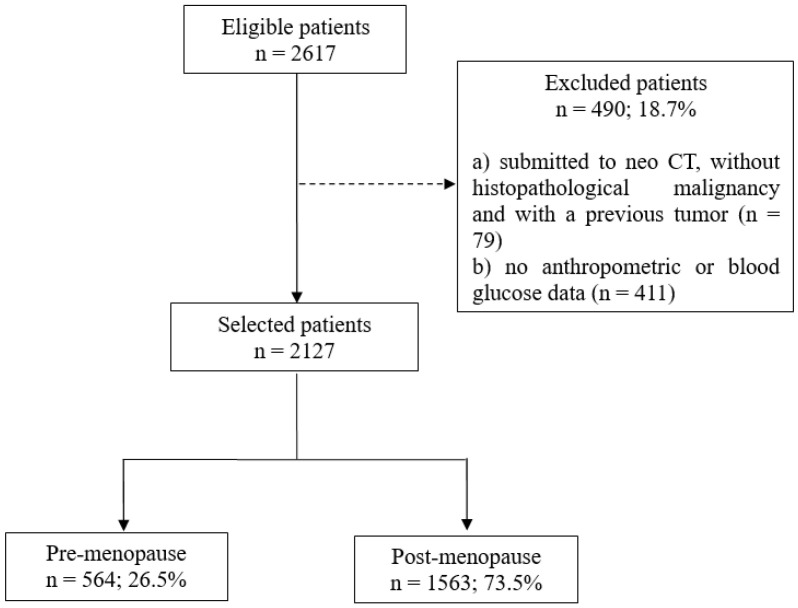
Patient selection flowchart. CT = chemotherapy.

**Figure 2 nutrients-14-04926-f002:**
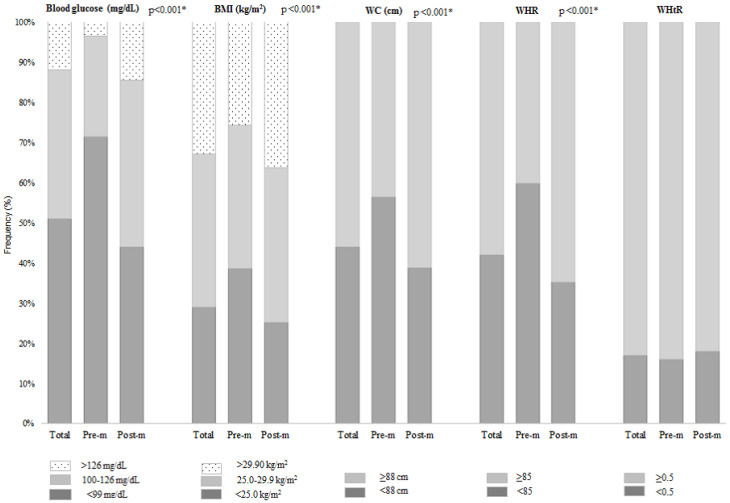
Fasting glucose, body mass index, body adiposity, and waist-to-height ratio at diagnosis in surgical patients with breast cancer according to menopausal period (*n* = 2.127). BMI = body mass index; WC = waist circumference; WHR = waist-to-hip circumference ratio; WHtR = waist-to-height ratio; Pre-m = premenopausal; Post-m = postmenopausal. * *p*-value refers to the chi-square test for proportions.

**Figure 3 nutrients-14-04926-f003:**
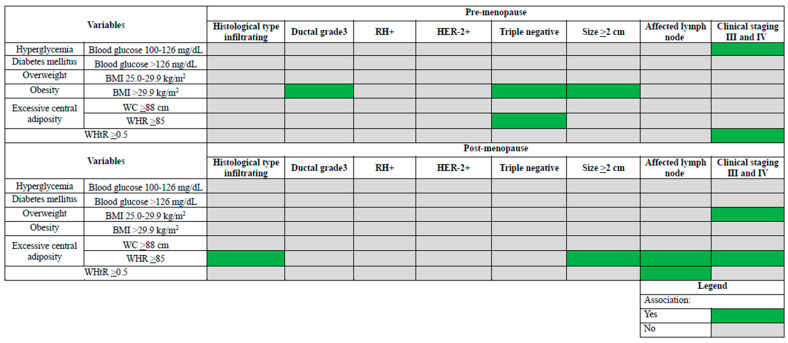
Synthesis of different factors associated with tumor characteristics in surgical patients with breast cancer according to menopausal status. BMI = body mass index; WC = waist circumference; WHR = waist-to-hip circumference ratio; WHtR = waist-to-height ratio; RH = hormone receptor; HER-2 = human epidermal growth factor receptor 2.

**Table 1 nutrients-14-04926-t001:** Characteristics of premenopausal breast cancer patients according to tumor characteristics (*n* = 564).

Variables	Total *n* (%)	Histological Type Infiltrating		Ductal Grade 3 ^b^		RH+ ^b^		HER-2+ ^b^		Triple Negative ^b^		Size ≥ 2 cm ^b^		Affected Lymph Node ^b^		Clinical Staging III and IV ^b^	
		No 93 (16.5%)	Yes 471 (83.5%)	No 216 (38.4%)	Yes 346 (61.6%)	No 87 (16.6%)	Yes 438 (83.4%)	No 326 (77.8%)	Yes 93 (22.2%)	No 376 (89.7%)	Yes 43 (10.3%)	No 188 (34.6%)	Yes 355 (65.4%)	No 311 (60.6%)	Yes 202 (39.4%)	No 434 (89.1%)	Yes 53 (10.9%)
Age (years)																	
<60	551 (97.7%)	91 (97.8%)	460 (97.7%)	211 (97.7%)	338 (97.7%)	86 (98.8%)	427 (97.5%)	316 (96.9%)	92 (98.9%)	366 (97.3%)	42 (97.7%)	184 (97.9%)	347 (97.7%)	231 (74.3%)	137 (67.8%)	428 (98.6%)	49 (92.4%)
≥60	13 (2.3%)	2 (2.2%)	11 (2.3%)	5 (2.3%)	8 (92.3%)	1 (1.2%)	11 (2.5%)	10 (3.1%)	1 (1.1%)	10 (2.7%)	1 (2.3%)	4 (2.1%)	8 (2.2%)	80 (25.7%)	65 (32.2%)	6 (1.4%)	4 (4.6%)
Blood glucose (mg/dL)																	
≤99	403 (71.4%)	70 (75.3%)	333 (70.7%)	151 (69.9%)	250 (72.3%)	64 (73.6%)	310 (70.8%)	232 (71.2%)	68 (73.1%)	270 (71.8%)	30 (69.8%)	138 (73.4%)	250 (70.4%)	228 (73.3%)	139 (68.8%)	316 (73.5%)	29 (54.7%) ^a^
100–126	141 (25.0%)	18 (19.3%)	123 (26.1%)	58 (27.3%)	82 (23.7%)	20 (23.0%)	113 (25.8%)	84 (25.8%)	22 (23.7%)	94 (25.0%)	12 (27.9%)	42 (22.3%)	94 (26.5%)	71 (22.8%)	57 (28.2%)	100 (23.0%)	22 (41.5%)
>126	20 (3.6%)	5 (5.4%)	15 (3.2%)	6 (2.8%)	14 (4.0%)	3 (3.4%)	15 (3.4%)	10 (3.0%)	3 (3.2%)	12 (3.2%)	1 (2.3%)	8 (4.3%)	11 (3.1%)	12 (3.9%)	6 (3.0%)	15 (3.5%)	2 (3.8%)
BMI (kg/m^2^)																	
≤24.9	218 (38.7%)	31 (33.3%)	187 (39.7%)	94 (43.5%)	124 (35.8%) ^a^	33 (38.0%)	173 (39.5%)	130 (39.9%)	30 (32.3%)	145 (38.6%)	15 (34.9%) ^a^	82 (43.6%)	127 (35.8%) ^a^	118 (37.9%)	80 (39.6%)	177 (40.8%)	16 (30.2%) ^a^
25.0–29.9	201 (35.6%)	37 (39.8%)	164 (34.8%)	78 (36.1%)	123 (35.6%)	27 (31.0%)	160 (36.5%)	112 (34.3%)	41 (44.1%)	142 (37.8%)	11 (25.6%)	71 (39.4%)	119 (33.5%)	119 (38.3%)	68 (33.7%)	154 (35.5%)	21 (39.6%)
>29.9	145 (25.7%)	25 (26.9%)	120 (25.5%)	44 (20.4%)	99 (28.6%)	27 (31.0%	105 (24.0%)	84 (25.8%)	22 (23.7%)	89 (23.6%)	17 (39.5%)	32 (17.0%)	109 (30.7%)	74 (23.8%)	54 (26.7%)	103 (23.7%)	16 (30.2%)
WC (cm)																	
<88	318 (56.4%)	52 (55.9%)	266 (56.5%)	129 (59.7%)	189 (54.6%)	47 (54.0%)	251 (57.3%)	189 (58.0%)	48 (51.6%)	215 (57.2%)	22 (51.2%)	122 (64.9%)	182 (51.3%) ^a^	181 (58.2%)	112 (55.4%)	255 (58.8%)	28 (52.8%)
≥88	246 (43.6%)	41 (44.1%)	205 (43.5%)	87 (40.3%)	157 (45.4%)	40 (46.0%)	187 (42.7%)	137 (42.0%)	45 (48.4%)	161 (42.8%)	21 (48.8%)	66 (35.1%)	173 (48.7%)	130 (41.8%)	90 (40.6%)	179 (41.2%)	25 (47.2%)
WHR																	
<85	338 (59.9%)	55 (59.1%)	283 (60.1%)	129 (59.7%)	208 (60.1%)	57 (65.5%)	259 (59.1%)	200 (61.3%)	54 (58.1%)	224 (59.6%)	30 (69.8%)	120 (63.8%)	203 (57.2%)	187 (60.1%)	121 (59.9%)	268 (61.7%)	31 (58.5%)
≥85	226 (40.1%)	38 (40.9%)	188 (39.9%)	87 (40.3%)	138 (39.9%)	30 (34.5%)	179 (40.9%)	126 (38.7%)	39 (41.9%)	152 (40.4%)	13 (30.2%)	68 (36.2%)	152 (42.8%)	124 (39.9%)	81 (40.1%)	166 (38.3%)	22 (41.5%)
WHtR																	
<0.5	88 (15.6%)	16 (17.2%)	72 (15.3%)	32 (14.8%)	56 (16.2%)	14 (16.1%)	66 (15.1%)	52 (15.9%)	12 (12.9%)	54 (14.4%)	10 (23.3%)	30 (16.0%)	53 (14.9%)	52 (16.7%)	27 (13.4%)	74 (17.0%)	4 (7.6%) ^a^
≥0.5	476 (84.4%)	77 (82.8%)	399 (84.7%)	184 (85.2%)	290 (83.8%)	73 (83.9%)	372 (84.9%)	274 (84.1%)	81 (87.1%)	322 (85.6%)	33 (76.7%)	158 (54.0%)	302 (85.1%)	259 (83.3%)	175 (86.6%)	360 (83.0%)	49 (92.4%)

Note: BMI = body mass index; WC = waist circumference; WHR = waist-to-hip circumference ratio; WHtR = waist-to-height ratio; RH = hormone receptor; HER-2 = human epidermal growth factor receptor 2. *p*-value refers to the chi-square test for proportions or Fisher’s exact test, with ^a^
*p*-value < 0.050. ^b^ variables with missing data.

**Table 2 nutrients-14-04926-t002:** Characteristics of postmenopausal breast cancer patients according to tumor characteristics (*n* = 1563).

Variables	Total *n* (%)	Histological Type Infiltrating		Ductal Grade 3 ^b^		RH+ ^b^		HER-2+ ^b^		Triple Negative ^b^		Size ≥ 2 cm ^b^		Affected Lymph Node ^b^		Clinical Staging III and IV ^b^	
		No 272 (17.4%)	Yes 1291 (82.6%)	No 589 (38.4%)	Yes 945 (61.6%)	No 283 (19.5%)	Yes 1167 (80.5%)	No 915 (81.8%)	Yes 203 (18.2%)	No 983 (87.9%)	Yes 135 (12.1%)	No 573 (37.8%)	Yes 943 (62.2%)	No 844 (60.0%)	Yes 563 (40.0%)	No 1179 (88.6%)	Yes 151 (11.4%)
Age (years)																	
<60	596 (38.1%)	101 (37.1%)	495 (38.3%)	231 (39.2%)	355 (37.6%)	104 (36.8%)	436 (37.4%)	371 (40.6%)	85 (41.9%)	398 (40.5%)	58 (43.0%)	231 (40.3%)	342 (36.3%)	304 (36.0%)	219 (38.9%)	452 (38.3%)	47 (31.1%)
≥60	967(61.9%)	171 (62.9%)	796 (61.7%)	358 (60.8%)	590 (62.4%)	179 (63.2%)	731 (62.4%)	544 (59.4%)	118 (58.1%)	585 (59.5%)	77 (57.0%)	342 (59.7%)	601 (63.7%)	540 (4.0%)	344 (61.1%)	727 (61.7%)	104 (68.9%)
Blood glucose (mg/dL)																	
≤99	689 (44.1%)	124 (45.6%)	565 (43.8%)	244 (41.4%)	432 (45.7%)	135 (47.7%)	502 (43.0%)	401 (43.8%)	91 (44.8%)	425 (43.2%)	67 (49.6%)	265 (46.2%)	392 (41.6%)	380 (45.0%)	227 (40.3%)	512 (43.4%)	71 (47.0%)
100–126	646 (41.3%)	109 (40.1%)	537 (41.6%)	257 (43.6%)	378 (40.0%)	107 (37.8%)	492 (42.2%)	379 (41.4%)	80 (39.4%)	405 (41.2%)	54 (40.0%)	232 (40.5%)	406 (43.0%)	341 (40.4%)	248 (44.1%)	496 (42.1%)	52 (34.5%)
>126	228(15.6%)	39 (14.3%)	189 (14.6%)	88 (15.0%)	135 (14.3%)	41 (14.5%)	173 (14.8%)	135 (14.8%)	32 (15.8%)	153 (15.6%)	14 (10.4%)	76 (13.3%)	145 (15.4%)	123 (14.6%)	88 (15.6%)	171 (14.5%)	28 (18.5%)
BMI (kg/m^2^)																	
≤24.9	394 (25.2%)	72 (26.5%)	322 (24.9%)	142 (24.1%)	243 (25.7%) ^a^	73 (25.8%)	296 (25.4%)	226 (24.7%)	53 (26.1%)	246 (25.0%)	33 (24.4%) ^a^	143 (25.0%)	236 (25.0%)	211 (25.0%)	143 (25.4%)	289 (24.5%)	48 (31.8%) ^a^
25.0–29.9	601 (38.5%)	107 (39.3%)	494 (38.3%)	208 (35.3%)	378 (40.0%)	108 (38.2%)	442 (37.9%)	346 (37.8%)	79 (38.9%)	369 (37.5%)	56 (41.5%)	232 (40.5%)	352 (37.3%)	342 (40.5%)	200 (35.5%)	471 (40.0%)	26 (26.5%)
>29.9	568 (36.3%)	93 (34.2%)	475 (36.8%)	239 (40.6%)	324 (34.3%)	102 (36.0%)	429 (36.7%)	343 (37.5%)	71 (35.0%)	368 (37.5%)	46 (54.1%)	198 (34.5%)	355 (37.7%)	291 (34.5%)	220 (39.1%)	419 (35.5%)	63 (41.7%)
WC (cm)																	
<88	607 (38.8%)	111 (40.8%)	496 (38.4%)	212 (36.0%)	382 (40.4%)	109 (38.5%)	450 (38.6%)	348 (38.0%)	79 (38.9%)	371 (37.7%)	56 (41.5%)	233 (40.7%)	353 (37.4%)	335 (39.7%)	212 (37.7%)	456 (38.7%)	65 (43.0%)
≥88	956 (61.2%)	161 (59.2%)	795 (61.6%)	377 (64.0%)	563 (59.6%)	174 (61.5%)	717 (61.4%)	567 (62.0%)	124 (61.1%)	612 (62.3%)	79 (58.5%)	340 (59.3%)	590 (62.6%)	509 (60.3%)	351 (62.3%)	723 (61.3%)	86 (57.0%)
WHR																	
<85	550 (35.2%)	112 (41.2%)^a^	438 (33.9%)	199 (33.8%)	340 (36.0%)	102 (36.0%)	397 (34.0%)	333 (36.4%)	66 (32.5%)	344 (35.0%)	55 (40.7%)	227 (39.6%)	304 (32.2%) ^a^	308 (36.5%)	177 (31.4%) ^a^	421 (35.7%)	36 (23.8%) ^a^
≥85	1013 (64.8%)	160 (58.8%)	853 (66.1%)	390 (66.2%)	605 (64.0%)	191 (64.0%)	770 (66.0%)	582 (63.6%)	137 (67.5%)	639 (65.0%)	80 (59.3%)	346 (60.4%)	639 (67.8%)	536 (63.5%)	386 (68.6%)	758 (64.3%)	115 (76.2%)
WHtR																	
<0.5	278 (17.8%)	51 (18.7%)	227 (17.6%)	97 (16.5%)	177 (18.7%)	59 (20.8%)	203 (17.4%)	173 (18.9%)	34 (16.7%)	176 (17.9%)	31 (23.0%)	97 (16.9%)	174 (18.5%)	134 (15.9%)	115 (20.4%)	206 (17.5%)	30 (19.9%)
≥0.5	1285 (82.2%)	221 (81.3%)	1064 (82.4%)	492 (83.5%)	768 (81.3%)	224 (79.2%)	964 (82.6%)	742 (81.1%)	169 (83.3%)	807 (82.1%)	704 (77.0%)	476 (83.1%)	769 (81.5%)	710 (84.1%)	448 (79.6%)	973 (82.5%)	121 (80.1%)

Note: BMI = body mass index; WC = waist circumference; WHR = waist-to-hip circumference ratio; WHtR = waist-to-height ratio; RH = hormone receptor; HER-2 = human epidermal growth factor receptor 2. *p*-value refers to the chi-square test for proportions or Fisher’s exact test, with ^a^
*p*-value < 0.050. ^b^ variables with missing data.

**Table 3 nutrients-14-04926-t003:** Factors associated with tumor characteristics in premenopausal surgical patients with breast cancer.

Variables	Histological Type Infiltrating OR (95% CI)		Ductal Grade 3 OR (95% CI)		RH + OR (95% CI)		HER-2 + OR (95% CI)		Triple Negative OR (95% CI)		Size ≥ 2 cm OR (95% CI)		Affected Lymph Node OR (95% CI)		Clinical Staging III and IV OR (95% CI)	
	Uni	Mult ^c^	Uni	Mult ^d^	Uni	Mult ^e^	Uni	Mult ^f^	Uni	Mult ^g^	Uni	Mult ^h^	Uni	Mult ^i^	Uni	Mult ^j^
Age (years)																
<60	1.00	-	1.00	-	1.00	-	1.00	-	1.00	-	1.00	-	1.00	-	1.00	-
≥60	0.91 (0.20–4.22)	-	0.99 (0.33–3.09)	-	2.21 (0.28–7.38)	-	2.91 (0.37–2.04)	-	0.87 (0.11–6.98)	-	1.06 (0.31–3.57)	-	1.10 (0.34–3.52)	-	5.82 (1.59–21.35) ^a^	-
Blood glucose (mg/dL)																
≤99	1.00	-	1.00	-	1.00	-	1.00	-	1.00	-	1.00	-	1.00	-	1.00	1.00
100–126	0.72 (0.49–1.06)	-	0.83 (0.57–1.24)	-	1.16 (0.68–2.01)	-	1.12 (0.65–1.92)	-	1.15 (0.56–2.33)	-	1.23 (0.81–1.88)	-	1.32 (0.87–1.98)	-	2.42 (1.33–4.40) ^b^	2.37 (1.34–4.56) ^b^
>126	1.40 (0.53–3.71)	-	1.41 (0.53–3.74)	-	1.03 (0.29–3.67)	-	0.98 (0.26–3.65)	-	0.75 (0.09–5.97)	-	0.76 (0.30–1.93)	-	0.82 (0.31–2.35)	-	1.47 (0.32–6.73)	1.21 (0.25–5.94)
BMI (kg/m^2^)																
≤24.9	1.00	-	1.00	1.00	1.00	-	1.00	-	1.00	1.00	1.00	1.00	1.00	-	1.00	-
25.0–29.9	1.03 (0.49–1.09)	-	1.19 (0.81–1.77)	1.19 (0.81–1.77)	1.13 (0.65–1.96)	-	0.63 (0.37–1.07)	-	0.75 (0.33–1.69)	0.87 (0.37–1.98)	1.04 (0.69–1.55)	1.04 (0.69–1.55)	0.84 (0.56–1.27)	-	1.51 (0.76–2.99)	-
>29.9	1.27 (0.81–2.23)	-	1.70 (1.09–2.66) ^a^	1.70 (1.09–2.66) ^a^	0.74 (0.42–1.30)	-	0.88 (0.48–1.62)	-	1.85 (0.88–3.88)	1.37 (1.08–3.24) ^a^	2.20 (1.36–3.56) ^a^	2.20 (1.36–3.56) ^a^	1.07 (0.68–1.69)	-	1.71 (0.82–3.58)	-
WC (cm)																
<88	1.00	-	1.00	-	1.00	-	1.00	-	1.00	-	1.00	-	1.00	-	1.00	-
≥88	1.02 (0.65–1.99)	-	1.23 (0.87–1.74)	-	0.87 (0.55–1.39)	-	0.77 (0.49–1.23)	-	1.27 (0.68–2.40)	-	1.76 (1.22–2.53) ^a^	-	1.12 (0.78–1.60)	-	1.27 (0.72–2.25)	-
WHR																
<85	1.00	-	1.00	-	1.00	-	1.00	-	1.00	1.00	1.00	-	1.00	-	1.00	-
≥85	1.04 (0.66–1.63)	-	1.98 (0.69–1.39)	-	1.31 (0.81–2.12)	-	0.87 (0.55–1.39)	-	1.67 (0.32–1.26)	1.53 (1.05–1.98) ^a^	1.32 (0.92–1.90)	-	1.01 (0.70–1.45)	-	1.14 (0.64–2.04)	-
WHtR																
<0.5	1.00	-	1.00	-	1.00	-	1.00	-	1.00	-	1.00	-	1.00	-	1.00	1.00
≥0.5	0.87 (0.48–1.57)	-	0.90 (0.56–1.44)	-	1.08 (0.57–2.03)	-	0.78 (0.39–1.53)	-	0.55 (0.26–1.19)		1.08 (0.66–1.76)	-	1.30 (0.79–2.15)	-	2.52 (0.88–7.19)	3.50 (1.14–10.77) ^a^

Note: BMI = body mass index; WC = waist circumference; WHR = waist-to-hip circumference ratio; WHtR = waist-to-height ratio; RH = hormone receptor; HER-2 = human epidermal growth factor receptor 2; UNI = univariate; MULT = multiple. *p*-value refers to logistic regression, with ^a^
*p*-value < 0.050 and ^b^
*p*-value < 0.001. Variables selected for the multiple models: ^c^ none; ^d^ BMI; ^e^ none; ^f^ age and BMI; ^g^ BMI and WHR; ^h^ BMI, WC, and WHR; ^i^ blood glucose; ^j^ age, blood glucose, BMI, and WHtR.

**Table 4 nutrients-14-04926-t004:** Factors associated with tumor characteristics in postmenopausal surgical patients with breast cancer.

Variables	Histological Type Infiltrating OR (95% CI)		Ductal Grade 3 OR (95% CI)		RH + OR (95% CI)		HER-2 + OR (95% CI)		Triple Negative OR (95% CI)		Size ≥ 2 cm OR (95% CI)		Affected Lymph Node OR (95% CI)		Clinical Staging III and IV OR (95% CI)	
	Uni	Mult ^c^	Uni	Mult ^d^	Uni	Mult ^e^	Uni	Mult ^f^	Uni	Mult ^g^	Uni	Mult ^h^	Uni	Mult ^i^	Uni	Mult ^j^
Age (years)																
<60	1.00	-	1.00	-	1.00	-	1.00	-	1.00	-	1.00	-	1.00	-	1.00	-
≥60	1.05 (0.80–1.38)	-	1.07 (0.87–1.32)	-	0.97 (0.74–1.27)	-	1.06 (0.77–1.44)	-	0.90 (0.63–1.30)	-	1.19 (0.96–1.50)	-	0.88 (0.71–1.10)	-	1.37 (0.96–1.98)	-
Blood glucose (mg/dL)																
≤99	1.00	-	1.00	-	1.00	-	1.00	-	1.00	-	1.00	-	1.00	-	1.00	-
100–126	1.92 (0.70–2.23)	-	0.83 (0.66–1.04)	-	1.24 (0.93–1.64)	-	1.07 (0.77–1.50)	-	0.84 (0.58–1.24)	-	1.18 (0.94–1.48)	-	1.22 (0.96–1.53)	-	0.76 (0.52–1.10)	-
>126	1.94 (0.63–2.40)	-	0.86 (0.63–1.18)	-	1.13 (0.77–1.68)	-	0.96 (0.61–1.50)	-	1.58 (0.61–1.06)	-	1.29 (0.94–1.77)	-	1.20 (0.87–1.65)	-	1.18 (0.74–1.89)	-
BMI (kg/m^2^)																
≤24.9	1.00	-	1.00	-	1.00	-	1.00	-	1.00	-	1.00	-	1.00	-	1.00	1.00
25.0–29.9	1.96 (0.49–3.09)	-	1.06 (0.81–1.39)	-	1.01 (0.72–1.41)	-	1.03 (0.70–1.51)	-	1.13 (0.71–1.79)	-	0.92 (0.70–1.20)	-	0.86 (0.66–1.13)	-	1.51 (1.32–1.80) ^a^	1.44 (1.28–1.70) ^b^
>29.9	1.27 (0.81–2.23)	-	1.79 (0.96–2.03)	-	1.04 (0.74–1.45	-	1.13 (0.76–1.68)	-	0.93 (0.57–1.49)	-	1.09 (0.83–1.42)	-	1.11 (0.85–1.47)	-	1.90 (0.90–2.36)	1.73 (0.47–2.12)
WC (cm)																
<88	1.00	-	1.00	-	1.00	-	1.00	-	1.00	-	1.00	-	1.00	-	1.00	-
≥88	1.90 (0.69–2.18)	-	1.83 (1.27–2.02)	-	0.99 (0.76–1.30)	-	1.04 (0.76–1.42)	-	1.85 (0.59–2.23)	-	1.14 (0.93–1.42)	-	1.09 (0.87–1.36)	-	0.83 (0.59–1.17)	-
WHR																
<85	1.00	1.00	1.00	-	1.00	-	1.00	-	1.00	-	1.00	1.00	1.00	1.00	1.00	1.00
≥85	1.73 (1.56–1.95) ^a^	1.73 (1.56–1.95) ^a^	1.91 (0.73–2.12)	-	1.09 (0.83–1.43)	-	0.84 (0.61–1.16)	-	1. 87 (0.54–2.13)	-	1.38 (1.11–1.71) ^a^	1.38 (1.11–1.71) ^a^	1.25 (0.99–1.57)	1.24 (1.02–1.56) ^a^	1.77 (1.20–2.63) ^a^	1.76 (1.30–2.65) ^a^
WHtR																
<0.5	1.00	-	1.00	-	1.00	-	1.00	-	1.00	-	1.00	-	1.00	1.00	1.00	-
≥0.5	1.92 (0.66–2.29)	-	0.85 (0.65–1.12)	-	1.25 (0.90–1.73)	-	0.86 (0.58–1.29)	-	1.73 (0.47–1.13)	-	0.90 (0.69–1.18)	-	1.73 (1.06–1.96) ^a^	1.70 (1.06–1.97) ^a^	0.83 (0.56–1.31)	-

Note: BMI = body mass index; WC = waist circumference; WHR = waist-to-hip circumference ratio; WHtR = waist-to-height ratio; RH = hormone receptor; HER-2 = human epidermal growth factor receptor 2; UNI = univariate; MULT = multiple. *p*-value refers to logistic regression, with ^a^
*p*-value < 0.050 and ^b^
*p*-value < 0.001. Variables selected for the multiple models: ^c^ RCQ; ^d^ BMI and PC; ^e^ fasting blood glucose and WHtR; ^f^ None; ^g^ fasting glucose, WHR, and WHtR; ^h^ age, fasting glucose, and WHR; ^i^ fasting blood glucose, WHR, and WHtR; ^j^ age, blood glucose, BMI, and WHR.

## Data Availability

Not applicable.

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
