# Peer review of "Body Weight, Central Adiposity, and Fasting Hyperglycemia Are Associated with Tumor Characteristics in a Brazilian Cohort of Women with Breast Cancer"

_nutrients, 2022, doi:10.3390/nu14224926_

Round 1

Reviewer 1 Report

In this observational study, authors investigated the association of anthropometry and fasting glucose with tumor characteristics in a cohort of women with breast cancer in Brazil. The manuscript is well written and easy to comprehend. However, there some points that need to be revised:

1. The title of the manuscrpit should define the study population e.g. "Body weight, central adiposity, and fasting hyperglycemia are associated with tumor characteristics in a Brazilian cohort of women with breast cancer"

2. Abstract: Authos should define the type of the study in the abstract (i.e., retrospective cohort study).

3. Introduction:

Author should provide more information on the role of obesity in breast cancer prognosis and breast-cancer specfivic mortality, including the potential underlying mechanism.

4. Methods:

line 62: Where there any exclusion criteria regadring perfomance status, e.g. as assessed by ECOG Pefromance Status scale.

line 62: Where there any other exclusion criteria? e.g. following a diet regimen for body weight loss, medication treatment to lose weight, co-existance of other chronic (e.g. cardiometabolic) or acute disease?

line 85: Why didn't you estimate body composition i.e. body fat mass ?

line 85: Did you estimate dietary patterns and nutritional intake with FFQ or other food records?

Discussion:

As shown by large cohort studyies (e.g. PREDIMED trial), dietary patterns and nutritional intake of breast cancer patients with obesity play a pivotal role in understanding their nutritional status. Why didn't you include such information in your study? Please add this in your study limitations.

Conclusions: Authors should revise conclusions (and abstract as well) regarding healthy eating or nutritional assessment, as they did not assess dietary intake. 

Author Response

Response to Reviewer 1 Comments

Comments and Suggestions for Authors

In this observational study, authors investigated the association of anthropometry and fasting glucose with tumor characteristics in a cohort of women with breast cancer in Brazil. The manuscript is well written and easy to comprehend. However, there some points that need to be revised:

  1. The title of the manuscrpit should define the study population e.g. "Body weight, central adiposity, and fasting hyperglycemia are associated with tumor characteristics in a Brazilian cohort of women with breast cancer"

Answer: Thanks for the reviewer's comment. We have corrected the title.

  1. Abstract: Authors should define the type of the study in the abstract (i.e., retrospective cohort study).

Answer: We appreciate the comment. The information about the type of study was already present in the third line of the abstract.

  1. Introduction: Author should provide more information on the role of obesity in breast cancer prognosis and breast-cancer specfivic mortality, including the potential underlying mechanism.

Answer: We agree with the reviewer's comment. We had described something about the mechanisms in the discussion. But now, we also add in the introduction succinctly.

  1. Methods:

line 62: Where there any exclusion criteria regadring perfomance status, e.g. as assessed by ECOG Pefromance Status scale. Where there any other exclusion criteria? e.g. following a diet regimen for body weight loss, medication treatment to lose weight, co-existance of other chronic (e.g. cardiometabolic) or acute disease?

Answer: Patients undergoing neoadjuvant chemotherapy, with absence of malignancy after histopathological confirmation, a history of primary tumor prior to breast cancer, and missing anthropometric and fasting blood glucose data from the hospital records were excluded. We do not use other exclusion criteria. It should be noted that all patients were evaluated from the pre-treatment phase.

line 85: Why didn't you estimate body composition i.e. body fat mass ?

Answer: Unfortunately, due to the retrospective nature of the study, it was not possible to assess body composition with the aid of image technics, however, WC is a good indicator of central obesity and has the advantage of being accessible in clinical practice;

line 85: Did you estimate dietary patterns and nutritional intake with FFQ or other food records?

Answer: As the above mentioned regarding body composition, the retrospective study design prevented us to assess data that was not part of the institutional routine.

  1. Discussion: As shown by large cohort studyies (e.g. PREDIMED trial), dietary patterns and nutritional intake of breast cancer patients with obesity play a pivotal role in understanding their nutritional status. Why didn't you include such information in your study? Please add this in your study limitations.

Answer: We appreciate the comment and we add this in the limitations. As this was a retrospective study of data collection from medical records and the institutional routine does not provide for the recording of dietary intake in the medical records, it was not possible to obtain it.

  1. Conclusions: Authors should revise conclusions (and abstract as well) regarding healthy eating or nutritional assessment, as they did not assess dietary intake.

Answer: Thanks for the reviewer's comment. We have corrected the conclusion.

Reviewer 2 Report

The purpose of this retrospective cohort study of women with breast cancer is to evaluate the association between the degree of obesity, abdominal circumference, hyperglycaemia and diabetes mellitus with the risk and type of breast cancer.

The correlation between the amount and distribution of body fat is a known risk factor, as is the association between alcohol and breast cancer risk.

The novelty of this study is in the evaluation of different tumour states and the degree of malignancy of the tumour with body fat and the presence of prediabetes or diabetes.

The authors should elaborate on these aspects that may actually represent novelties compared to what the extensive literature has already shown.

Minor points:

The tables are difficult to read. They need to be improved.

Figure 1 is unclear. It must be made more readable.

error: "87 Central adiposity was measured using WC (< 0.88 cm)"

Author Response

ANSWERS TO THE REVIEWER 2 COMMENTS

Reviewer: 2

Comments and Suggestions for Authors

The purpose of this retrospective cohort study of women with breast cancer is to evaluate the association between the degree of obesity, abdominal circumference, hyperglycaemia and diabetes mellitus with the risk and type of breast cancer.

  1. The correlation between the amount and distribution of body fat is a known risk factor, as is the association between alcohol and breast cancer risk. The novelty of this study is in the evaluation of different tumour states and the degree of malignancy of the tumour with body fat and the presence of prediabetes or diabetes. The authors should elaborate on these aspects that may actually represent novelties compared to what the extensive literature has already shown.

Answer:  We appreciate the comment and we add these aspects in the article discussion.

Minor points:

  1. The tables are difficult to read. They need to be improved.

Answer: We agree with the reviewer's comment. Because the study involved different ways of assessing nutritional status and different tumor characteristics, the tables were really big. We understand that this can really make it difficult to read them, but it is a difficult situation to change. In this way, we inserted a color resource (gray tone) in the tables in an attempt to make it easier for the reader to visualize the results.

  1. Figure 1 is unclear. It must be made more readable.

Answer: We agree with the reviewer's comment. We believe that the reviewer is referring to figure 2, as this is the one that is not really clear or with good resolution. We make the figure clearer.

  1. error: "87 Central adiposity was measured using WC (< 0.88 cm)"

Answer: Thanks for the reviewer's comment. We have corrected this information.

Round 2

Reviewer 1 Report

Authors revised their manuscript accrording to suggested comments. The article is eligible for publication.

Reviewer 2 Report

The authors fixed all the issues I had raised.